# An Update on Clinical Trials and Potential Therapeutic Strategies in T-Cell Acute Lymphoblastic Leukemia

**DOI:** 10.3390/ijms24087201

**Published:** 2023-04-13

**Authors:** Janisha Patel, Xueliang Gao, Haizhen Wang

**Affiliations:** 1Department of Cell and Molecular Pharmacology & Experimental Therapeutics, Medical University of South Carolina, Charleston, SC 29425, USA; patejani@musc.edu (J.P.); gaox@musc.edu (X.G.); 2Hollings Cancer Center, Medical University of South Carolina, Charleston, SC 29425, USA; 3Department of Pediatric Hematology/Oncology, Medical University of South Carolina-Shawn Jenkins Children’s Hospital, Charleston, SC 29425, USA

**Keywords:** T-cell acute lymphoblastic leukemia, clinical trials, relapse, refractory, immunotherapy, CAR-T therapy

## Abstract

Current therapies for T-cell acute leukemia are based on risk stratification and have greatly improved the survival rate for patients, but mortality rates remain high owing to relapsed disease, therapy resistance, or treatment-related toxicities/infection. Patients with relapsed disease continue to have poor outcomes. In the past few years, newer agents have been investigated to optimize upfront therapies for higher-risk patients in the hopes of decreasing relapse rates. This review summarizes the progress of chemo/targeted therapies using Nelarabine/Bortezomib/CDK4/6 inhibitors for T-ALL in clinical trials and novel strategies to target NOTCH-induced T-ALL. We also outline immunotherapy clinical trials using monoclonal/bispecific T-cell engaging antibodies, anti-PD1/anti-PDL1 checkpoint inhibitors, and CAR-T for T-ALL therapy. Overall, pre-clinical studies and clinical trials showed that applying monoclonal antibodies or CAR-T for relapsed/refractory T-ALL therapy is promising. The combination of target therapy and immunotherapy may be a novel strategy for T-ALL treatment.

## 1. Introduction

T-cell acute lymphoblastic leukemia (T-ALL) compromises about 10–15% of newly diagnosed pediatric ALL cases and about 25% of adult ALL cases [1,2]. T cells develop in the thymus. The accumulation of genetic mutations and aberrant proliferation of immature progenitors result in T-ALL [1,3]. The overall survival rates are generally higher in the pediatric population compared with the adult population, with rates of 80% and 50%, respectively. Lower survival rates in the adult population are generally due to higher toxicities caused by treatment. Current therapies are based on risk stratification and have greatly improved the survival rate for patients, but mortality rates remain high owing to relapsed disease, therapy resistance, or treatment-related toxicities/infection [1]. Patients with relapsed disease continue to have poor outcomes, with event-free and overall survival rates of less than 25% [4]. The current therapeutic goal in relapsed disease is to achieve a subsequent remission and then treat with an allogeneic hematopoietic cell transplant to ensure a complete response [5]. Newer agents have been investigated in the past few years to optimize the therapies for higher-risk patients in order to decrease the relapse rates [4]. The chemo/targeted therapies and immunotherapy for relapsed/refractory T-ALL in clinical trials are summarized in this review. We outline the applications of Nelarabine, Bortezomib, and CDK4/6 inhibitors for T-ALL therapy in preclinical models and in clinical trials. For immunotherapy, monoclonal antibodies/bispecific T-cell engaging antibodies, anti-PD1/PDL1 checkpoint inhibitors, and CAR-T for T-ALL therapy are summarized.

## 2. Results

### 2.1. Chemotherapy/Targeted Therapy for T-ALL

The current standard for T-ALL care is based on risk stratification and response to therapy based on minimal residual disease (MRD). Prior clinical trials have established that early intensive and multi-agent chemotherapy improves T-ALL therapeutic outcomes [2]. Early intensification therapy includes a four-drug induction containing dexamethasone and anthracycline with intrathecal chemotherapy. This is followed by consolidation therapy, which contains the augmented Berlin–Frankfurt–Muenster regimen (Cyclophosphamide, Cytarabine, Vincristine, Mercaptopurine, and Pegaspargase) and, more recently, the addition of Nelarabine in higher risk patients based on clinical trial AALL0434 [6]. Depending on the response, additional chemotherapy is given with high-dose Methotrexate (HD-MTX) or dose-escalating Methotrexate (c-MTX), followed by maintenance chemotherapy [1,6].

#### 2.1.1. Nelarabine

Nelarabine is a chemotherapy drug recently added to the standard of therapy for treating T-ALL in higher-risk patients. Nelarabine is a purine nucleoside analog metabolized into arabinosylguanine nucleotide triphosphate, accumulating in T lymphoblasts and killing T-ALL cells [6]. Nelarabine was previously studied as a single agent in a phase I study (*n* = 93) and was found to have a significant cytotoxic activity against malignant T cells, with 54% of patients with T-ALL achieving a complete or partial response after one to two courses of the drug [7]. These results were replicated in a phase II trial (*n* = 121), which showed a 55% response rate in patients with leukemia in the first relapse [8]. Most recently, a phase III trial, AALL0434 (*n* = 1562), evaluating Nelarabine in newly diagnosed T-ALL, also showed promising results. In this trial, patients with intermediate-risk or high-risk T-ALL who either received or did not receive six 5-day courses of Nelarabine were incorporated into the augmented Berlin–Frankfurt–Muenster regimen. The 5-year leukemia-free survival was superior in patients who received Nelarabine, at 88.2% versus 82.1% (*p* = 0.029) in those who did not receive Nelarabine. Similarly, overall survival (OS) was better in patients who received Nelarabine, at 90.3% compared with those who did not at 87.9% (*p* = 0.168). Additionally, patients who received Nelarabine had fewer central nervous system (CNS) relapses and comparable toxicities to those patients who did not receive Nelarabine [6]. Of note, a recent report showed that Nelarabine induced myelopathy in patients after the transplantation of allogeneic hematopoietic cells [9]. Thus, it is critical to study the mechanism of Nelarabine-induced myelopathy in order to develop a better therapeutic strategy.

#### 2.1.2. Bortezomib

Bortezomib is another chemotherapy agent that has been recently investigated for treatment in T-ALL. It is a proteasome inhibitor, which results in the inhibition of pro-apoptotic factors [10]. The Children’s Oncology Group (COG) recently completed the AALL1231 phase III clinical trial (*n* = 824), which evaluated Bortezomib in newly diagnosed T-ALL. Children/young adults with T-ALL or T-lymphoblastic lymphoma were randomly assigned to a modified augmented Berlin–Frankfurt–Munster chemotherapy regimen with or without Bortezomib during induction and delayed intensification. Additionally, two extra doses of Pegaspargase and Dexamethasone were applied instead of Prednisone in this trial. The overall outcomes showed no statistical differences in the group of patients who received Bortezomib compared with those who did not. Compared with the AALL0434 trial (using Nelarabine), patients who received Bortezomib in the AALL1231 trial did not have significantly better outcomes. In the AALL1231 trial, patients who received Bortezomib had an event-free survival (EFS) of 90.1% and 4-year OS of 93.6%, compared with those patients who received Nelarabine in the AALL0434 trial with EFS of 86.6% (patients on the HD MTX arm) and 92.2% (patients on the C-MTX arm) and OS of 90.1% (HD MTX) and 92.9% (C-MTX) [11]. There does not seem to be a significant benefit of Bortezomib in T-ALL. Although the addition of Nelarabine has some beneficial prospects, there remains room for improvement.

#### 2.1.3. Targeting CDK4/6

Deregulation of cell cycle progression is a key characteristic of cancer cell proliferation. Cyclins and their associated kinases are the core machineries in regulating the cell cycle and cell proliferation [12]. Cyclin-D-dependent kinase 4 or 6 (CDK4/6) forms a complex with cyclin D, resulting in catalytic activity to promote cell cycle progression at the G1-S phase checkpoint. Numerous aberrations within this system lead to the loss of this proliferative control, including, but not limited to, overexpression of CDK4/6, amplification of cyclin D, and mutation/depletion of cyclin-dependent kinase inhibitors [13,14,15,16]. Our previous study showed that cyclin D3 and CDK6 are highly expressed in T-ALL [17]. The CDK6 kinase activity in T-ALL is enhanced as a result of the high mutation rate of its inhibitors, CDKN2A and CDKN2B (over 50% of the cases).

CDK4/6 inhibitors, such as Palbociclib and Ribociclib, have been developed as potential treatment options for T-cell ALL [17,18,19,20,21]. These agents are proposed to induce cell cycle arrest at the G1/S checkpoint, induce leukemia cell apoptosis, and prevent leukemia infiltration into multiple organs. Pre-clinical studies have demonstrated the efficacy of single-agent activity for CDK4/6 inhibitors, but combinations with other cytotoxic medications have not shown enhanced anti-neoplastic efficacy [22,23]. One pre-clinical study that evaluated the effects of this combination therapy on T-ALL cell lines showed promising results. They found that CDK4/6 inhibitors synergized with drugs such as Glucocorticoids, which do not rely on rapidly proliferating cells. The same effects were found on murine T-ALL models. Many chemotherapy agents are only effective on rapidly proliferating cells, which limits the application of CDK4/6 inhibitors in combination with those agents [24,25]. In another pre-clinical study, the CDK4/6 inhibitor markedly decreased the number of human CD45+ cells (a marker for T-ALL) in vivo in T-ALL patient-derived xenograft (PDX) models. This same study demonstrated that the administration of Palbociclib given concurrently with Vincristine led to a significant decrease in disease burden compared with the monotherapy [18]. Thus far, clinical trials investigating Palbociclib and other CDK4/6 inhibitors for T-ALL therapy have been limited. In one COG pilot study (*n* = 8), patients with refractory ALL or relapsed ALL were given Palbociclib once daily for 21 days as a single agent, followed by a combination with four-drug re-induction chemotherapy. The preliminary results have shown that this regimen is safe and well tolerated in children and young adults, but the final conclusion has not been made as this study is still ongoing [26]. Of note, our recent studies show that targeting CDK6 induces T-ALL apoptosis [17] and inhibits the infiltration/metastasis of leukemia into different organs [19]. This suggests that specifically targeting CDK6 is promising for T-ALL therapy.

#### 2.1.4. Targeting NOTCH Signaling

NOTCH signaling is highly conserved throughout the animal kingdom and promotes T-ALL cell proliferation. Over 50% of T-ALL has been identified with activation mutations in NOTCH1. This leads to a broad interest in exploring the mechanisms of oncogenic NOTCH in T-ALL initiation and progression [27]. However, targeting NOTCH signaling using γ-secretase inhibitors (GSIs) for T-ALL has been held back owing to the low antileukemic effect and the development of gastrointestinal toxicity [28]. Through exploring the NOTCH-related/induced molecular mechanisms, new therapeutic strategies targeting NOTCH signaling are under development, which hold promise for T-ALL treatment. The following are examples of recent studies targeting NOTCH-induced T-ALL. Epigenetic dysfunction plays a significant role in T-ALL development, and the histone lysine demethylase KDM6B is significantly overexpressed in T-ALL. A study showed that KDM6B is required for the development and maintenance of T-ALL. KDM6B deficiency in murine cells was not able to develop T-ALL in a NOTCH1-based retroviral mouse model. Meanwhile, KDM6B is required to prevent apoptosis of NOTCH1-driven T-ALL. Therefore, targeting KDM6B might be an efficient therapeutic strategy for NOTCH1-induced T-ALL patients [29]. Other studies showed that transgenic deletion of interleukin-15 (IL-15) or IL-15 receptor alpha-chain generates T-ALL in NOD mice accompanied by increased NOTCH1 activation and impaired DNA repair capability. This suggests that IL-15 may be applied to prevent T-ALL development [30]. A study showed that cell division cycle 73 (Cdc73), a NOTCH cofactor, plays a critical role in protecting NOTCH-induced T-ALL from mitochondria stress and DNA damage. Targeting Cdc73 may weaken NOTCH signals without directly targeting NOTCH [31]. On the other hand, circular RNAs (circRNAs) are potentially critical in regulating leukemia development. CircFBXW7 is highly expressed in T cells, and circFBXW7 depletion increases T-ALL proliferation and cell viability by increasing MYC and NOTCH protein levels [32]. Thus, circFBXW7 may be applied as a therapeutic strategy for T-ALL treatment. In addition, a recent study showed that extracellular vesicle microRNAs might contribute to the NOTCH signaling pathway by acting as autocrine stimuli in T-ALL. Therefore, targeting extracellular vesicle microRNAs may also be a novel therapeutic strategy for T-ALL treatment [33].

### 2.2. Immunotherapy for Relapsed/Refractory T-ALL

Immunotherapies, including monoclonal antibodies, bispecific T-cell-engaging antibodies, and chimeric antigen receptor (CAR)-modified T cells, have been successfully implemented in treating B-cell acute lymphoblastic leukemia (B-ALL). However, only a few immunotherapies have been studied in T-ALL. This continues to be difficult as it requires a target unique to T-ALL cells, but does not present on normal T lymphocytes. Targeting specific molecules on T-ALL may avoid toxicity related to prolonged T-cell depletion [4]. Of note, some monoclonal antibodies may warrant the investigation.

#### 2.2.1. Monoclonal Antibodies/Bispecific T-Cell-Engaging Antibodies for T-ALL Therapy

##### IL-7 Ralpha Monoclonal Antibodies

Interleukin-7 receptor alpha subunit (IL-7Rα, CD127) is a component of the receptor for two distinct cytokines, IL-7 and IL-7 homolog (thymic stromal lymphopoietin). The binding of IL-7 to IL-7Rα activates the RAS/MAPK/ERK, PI3K, and JAK-STAT pathways to function in the development/proliferation/homeostasis of T cells, B cells, and NK cells [34,35,36]. IL-7Rα is upregulated in relapsed T-ALL and is a potential target for T-ALL treatment. Applying anti-IL-7Ra monoclonal antibodies for patient-derived xenografts derived from relapsed T-ALL can effectively control the disease progression [37]. Another study showed that anti-IL-7Ra antibodies can induce natural-killer-mediated T-ALL cell death in vitro and inhibit T-cell leukemia progression in vivo [38]. However, normal T cells also express IL-7Ra. Thus, applying anti-IL-7Ra antibodies may cause T-cell depletion, which impedes the application of anti-IL-7Ra antibodies for clinical trials.

##### Alemtuzumab

CD52 is a glycoprotein anchored to glycosylphosphatidylinositol and is expressed on the cellular surface of multiple immune cells, including mature lymphocytes, monocytes/macrophages, natural killer cells, and dendritic cells. The ligation of CD52 may function in T-cell activation and proliferation by offering costimulatory signals [39]. Alemtuzumab is an anti-CD52 humanized monoclonal antibody and induces killing by complement-dependent cytotoxicity, antibody-dependent cellular cytotoxicity, and apoptosis of target cells [40]. Several clinical trials have been conducted to test its efficacy for multiple subtypes of T-cell malignancies. In one trial (*n* = 24), adult patients with B- and T-ALL who presented with more than 10% of CD52 positive lymphocytes were given Alemtuzumab combined with chemotherapeutic drugs. In these patients, MRD was reduced, but significant toxicity was noted with viral reactivation of cytomegalovirus, herpes simplex, or herpes zoster latent infections. This ultimately prompted the termination of this trial [41]. In addition, Alemtuzumab did not show promising results as a single agent in a phase II clinical study. It showed improved MRD when combined with G-CSF [42]. After three clinical trials, no additional trials have been initiated for T-ALL with Alemtuzumab so far (Table 1).

##### Daratumumab

CD38 is a transmembrane glycoprotein that functions in cell migration, cell adhesion, and signal transduction. Daratumumab, a monoclonal antibody that targets CD38, has also gained attention over the last few years. Although Daratumumab was initially developed for multiple myeloma, it is a promising target for T-ALL. CD38 is expressed on T-ALL and early T-cell precursor (ETP)-ALL blasts during diagnosis, chemotherapy treatment, and relapse [1]. Additionally, CD38 is expressed at low levels in normal lymphoid and myeloid cells. Pre-clinical data suggest that CD38 expression is not downregulated by conventional cytotoxic chemotherapy. Using T-ALL PDX models, Daratumumab administration significantly reduced the leukemia burden in 14 out of 15 PDX models [43]. The initial clinical trials investigating Daratumumab showed promising results. In the phase II DELPHINUS study (*n* = 47), Daratumumab was given with vincristine, prednisone, PEG-asparaginase, and doxorubicin (VPLD) in cycle 1 and with methotrexate, cyclophosphamide, cytarabine, and 6-mercaptopurine in cycle 2 in patients with first relapse or refractory T-ALL. Adding Daratumumab improved response rates compared with those patients who received chemotherapy only [44]. There is currently an ongoing clinical trial in phase II using anti-CD38 to treat T-ALL with persistent or recurrent MRD after chemotherapy treatment (Table 1).

##### CD3–CD38 Bispecific Antibodies

As a transmembrane glycoprotein, CD38 is expressed in T-ALL and acute myeloid leukemia (AML) cells [45,46]. XmAb18968, a CD3–CD38 bispecific antibody, has optimized relative binding capability for CD3 and CD38. The binding of CD3 and CD38 leads to reduced cytokine release, while the killing capability on target cells is not affected. Applying the CD3–CD38 bispecific antibody for treating T-ALL, T-cell lymphoblastic lymphoma, or AML in the relapsed/refractory status is under phase I clinical trial (NCT05038644) [47].

#### 2.2.2. CAR-T for T-Cell Malignancy

CAR-T-cell therapy has been thoroughly studied and has shown success in treating B-cell leukemia. However, the applications of CAR-T for T-ALL treatment are much fewer [48]. To generate CAR-T cells, T cells are collected via aphaeresis and, in the laboratory, T cells are genetically engineered by inserting genes into the cells to produce chimeric antigen receptors (CARs) on their cell surfaces. These CAR-T cells are then infused into the patient, typically after chemotherapy. Challenges, such as T-cell aplasia and no tumor-specific antigens, arise when applying this therapy to T-cell malignancies. With the identification of potential targets, the therapeutic efficacy of CAR-T in T-cell malignancies is currently being evaluated in several phase I and/or II clinical trials [48].

##### CAR-T in Clinical Trials and Preclinical Studies

The current phase I and phase II clinical trials regarding CAR-T for T-cell malignancy are summarized in Table 2. CD7, a transmembrane glycoprotein, is expressed by T cells, natural killer cells, and their precursors. It is also ubiquitously expressed in T-cell leukemia/lymphoma and some peripheral T-cell lymphoma [49,50,51]. CD7 is being examined as a potential target in CAR-T [49]. A single-center, early-phase clinical trial showed impressive therapeutic efficacy of anti-CD7 CAR-T cells for relapsed and/or refractory T-ALL (ChiCTR2000034762) [52]. In this trial, anti-CD7 CAR-T cells were administered to each participant (*n* = 20, all of whom had received at least two previous therapies). Overall, the infusions were well tolerated with no dose-limiting toxicities and all reversible adverse events, with the exception of one patient. Regarding efficacy, 90% of patients (*n* = 18) achieved complete remission. Of these patients, six remained in remission after additional therapy with stem cell transplant and nine remained in remission at the 7-month follow-up [53]. So far, there are seventeen phase I/II clinical trials using CAR-T to target CD7 for relapsed/refractory T-cell leukemia/lymphoma/malignancy therapy.

CD4 is universally expressed in peripheral T-cell lymphoma cells. Recent studies suggest that CD4CAR can eliminate CD4-positive T-ALL in a mouse model. Mice injected with CD4CART had a 50% reduced tumor burden than control mice on days 3, 6, and 8. These data suggest that the application of CD4CAR as a therapy against CD4-positive T-cell malignancies is promising [54]. One phase I clinical trial (NCT03829540) uses CAR-T to target CD4 for T-cell leukemia/lymphoma.

CD5 is another highly expressed marker on malignant T cells, but not on hematopoietic stem cells [55]. CD5 CAR-T cells have been shown to effectively target various T-cell cancers in preclinical studies [55,56,57]. Importantly, CD5 CAR-T cells specifically target malignant T cells without affecting the normal T cell population [56,57]. In one single patient study, CD5 CAR-T cells were modified to secrete IL-15/IL-15 sushi complex, as IL-15 has been shown to strengthen the anti-tumor response. These modified cells were administered to a patient with relapsed T-lymphoblastic lymphoma accompanied by CNS involvement and successfully resulted in the remission of the disease [55]. Although more investigation is needed, these data showed promising prospects for CD5 CAR-T. Four clinical trials (phase I) use CAR-T to target CD5 for T-cell leukemia/lymphoma/T-cell malignancy therapy.

TRBC1 and TRBC2 are the two TCR beta chain constant regions. Normal T cells have a mutually exclusive expression of either TRBC1 or TRBC2. Strategies to target TRBC1 kill the TRBC1+ T-cell malignancy and TRBC1+ normal T cells, but keep a substantial proportion of the TRBC2+ normal T-cell compartment intact to maintain cellular immunity [58]. A clinical trial (phase I) is using anti-TRBC1 CAR-T-cell therapy for T-cell malignancy (Table 2).

Cortical T-ALL (coT-ALL) is a major subset of T-ALL. CD1a, a cell surface antigen, is specifically expressed in coT-ALL and retained in relapsed coT-ALL, but barely expressed in normal cells/tissues. CD1a-specific CAR-T therapy has been validated preclinically for relapsed/refractory coT-ALL [59]. A recent phase II clinical trial of anti-CD1a CAR-T in treating R/R T-ALL was initiated (NCT05745181). Meanwhile, it is challenging to achieve sufficient effector T cells from relapsed/refractory patients owing to the aggressiveness and hyper leukocytic of the disease. A study suggests that engineered T cells secreting CD1a XCD3 T-cell-engaging antibodies (CD1a-STAb) may be applied for coT-ALL patients with limited numbers of effector T cells [59]. This study showed that CD1a-T-cell engagers induce specific T-cell activation through binding to CD1a and CD3 on the cell surface. Moreover, the cytotoxicity effect induced by CD1a-STAbs is better than that of CD1a-CAR-T cells when the effector-to-target ratio is low.

CD99, a glycosylated transmembrane protein, has multiple cellular functions, such as cell death, cell differentiation, cell adhesion/migration, endocytosis, exocytosis, and intracellular protein trafficking. CD99 is highly expressed in most T-ALL. CD99-based CAR T cells eradicate T-ALL efficiently with no apparent toxicity on normal blood cells [60]. This suggests that applying CD99-based CAR-T cells for T-ALL therapy is a promising strategy. Currently, two multi-institutional phase I studies are undergoing to evaluate the efficiency and toxicity of anti-CD99 CAR T cells (ChiCTR2100046764, ChiCTR2000033989).

Pre-clinical studies of CAR-T: Leukemia cells from individual patients have a unique cellular surface TCR, which can distinguish leukemia cells from normal T cells. One study showed that targeting the unique complementarity-determining region 3 (CDR3) on TCRs on leukemia cells using CDR3-selective CAR T cells is promising for leukemia therapy, including T-ALL [61]. CD21 is expressed in 70% of human T-ALL cell lines and 57% of primary T-ALL samples. While the expression of CD21 in mature T cells is very low, CD21 is expressed in 80% of cortical T-ALL, 72% of pre-T, and 67% of mature T cells. Importantly, the anti-CD21 CAR-T cell is efficient for the murine model of T-ALL [62].

No phase III and phase IV clinical trials were disclosed for T-ALL therapy using CAR-T in general by the end of the year 2022. This suggests that CAR-T-cell therapy for T-ALL is promising, but more work needs to be done before the conclusion about the efficacy of the therapy on T-ALL is made.

##### Other CAR Therapies

CAR-T cells and their target leukemia may express the same antigen, which leads to the CAR-T cell fratricide. This is serious in CAR-T cell therapy for T-ALL [63]. On the other hand, studies showed that NK cells, exhibiting a strong cytolytic function against tumor cells, do not share most of the antigens targeted for T-cell malignancy. Preclinical studies showed that CAR-modified NK cells effectively suppress the progress of T-cell malignancies [64,65,66]. Preclinical and clinical trials using CAR-NK cells (CD7 CAR-NK, CD5 CAR-NK, and CD4 CAR-NK) for treating CD7-/CD5-/CD4-positive T-cell malignancy have been initiated [67]. However, the CAR-NK application displayed certain limitations owing to the difficulties in expanding and transduction with viral vectors, which led to the development of NK-92-based CAR products. To prevent permanent engraftment of NK-92 cells, irradiation is required before the infusion of NK-92-based CAR cells [68,69]. On the other hand, a promising CAR macrophage trial has been applied for solid tumors (NCT04660929). Future studies may test whether the CAR-macrophage also works for T-ALL.

#### 2.2.3. Anti-PD-1/PD-L1 Abs

Finally, we would like to discuss the potential implications of immunotherapy through checkpoint inhibitors. The PD-1 pathway is important in T-cell regulation. It is expressed on tumor-specific activated T cells, and its binding to PD-L1 on tumor cells inhibits the cytotoxic T-cell-mediated tumor cell killing. As a result, PD-1/PD-L1 inhibition should reactivate anti-tumor responses by reducing the activity of regular T cells and restoring the activity of effector T cells. PD-1 and PD-L1 monoclonal antibodies have shown promising effects in myeloid, lymphoid, and virus-related hematological malignancies [70]. Patients with NK-/T-cell lymphoma who have not responded to L-asparaginase-containing regimens have also responded well to the treatment of anti-PD1 antibody pembrolizumab or nivolumab [71,72]. Nivolumab was studied in a phase II clinical trial in patients with T-ALL; unfortunately, after the first three patients enrolled, the drug causing the progression of the disease was found (NCT02631746) [73]. Given the discrepancy, further investigations of PD-1 and PD-L1 monoclonal antibodies for T-ALL patients’ therapy are certainly needed. Interestingly, a recent study showed that PD-1-expressing leukemia stem cells initiate disease as those cells have high NOTCH1-MYC activity. Meanwhile, PD-1 signaling protects cells from the T-cell-receptor-signal-induced apoptosis [74]. This study suggests that targeting PD-1-expressing leukemia stem cells by PD-1 blockade may be a promising strategy for T-ALL therapy.

## 3. Conclusions

Outcomes in T-ALL patients have significantly improved over the last few decades. However, patients with refractory and relapsed diseases continue to have poor outcomes. This review discusses the clinical trials completed in the last decade and the prospective immunotherapy options. Overall, immunotherapy against T-ALL is mainly at the early clinical research stage or under preclinical development. Developing antibodies to target cell surface receptors is a promising therapeutic strategy for T-ALL, while the major limitation is that immune cells, such as normal T cells, may also express the corresponding receptors. Thus, applying these antibodies will cause side effects, such as T-cell depletion. On the other hand, even though anti-PD-1/PD-L1 antibodies are showing certain success in treating solid tumors, applying these antibodies may block the PD-1 pathway in T-ALL and lead to tumor progression. Thus, the application of checkpoint inhibitors for T-ALL therapy needs to be further evaluated. The promising CAR-T therapy for T-ALL is still at an early stage, though CAR-T therapy for diffuse large B-cell lymphoma or mantle cell lymphoma has achieved success. Identifying T-ALL-specific targets for immunotherapy remains challenging, but several preclinical studies showed promise in investigating monoclonal antibodies and CAR-T for relapsed/refractory T-ALL. Novel strategies of CAR-T on T-ALL therapy may need to avoid cell fratricide, tumor cell contamination, and T-cell aplasia. In addition, because certain target therapies can augment the immunotherapy efficacy in solid tumors, it is worth testing the combinational therapeutic efficacy of target therapy with immunotherapy for T-ALL patients.

## Figures and Tables

**Table 1 ijms-24-07201-t001:** Clinical trials using monoclonal/bispecific antibodies for the treatment of relapsed/refractory T-ALL.

Target	Clinical Trials	Phase	Condition	Age	Location	Status
CD38	NCT05289687	II	T-ALL with persistent or recurrent MRD after chemotherapy treatment	18 years and older	Northwestern, Chicago, Illinois, United States	Recruiting
CD52	NCT02689453	I	T-cell Lymphoma Relapsed Adult T-Cell Leukemia (ATL) Peripheral T-Cell Lymphoma (PTCL) Cutaneous T-Cell Lymphoma (CTCL) T-Cell Prolymphocytic Leukemia	18 years and older	National Institutes of Health Clinical Center, Maryland, United States	Completed
CD52	NCT00061048	II	Acute T-Cell Leukemia-Lymphoma	18 years and older	National Institutes of Health, Maryland, United States	Completed
CD52	NCT00199030	II	R/R Adult Acute Lymphocytic Leukemia T-cell Lymphoma, Lymphoblastic	18 years and older	University Hospital Frankfurt, Germany	Completed
CD3–CD38	NCT05038644	I	R/R T-Cell Acute Lymphoblastic Leukemia/Acute Myeloid Leukemia/T-Cell Lymphoblastic Lymphoma	18 years and older	Mayo Clinic, Arizona Mayo Clinic, Florida Moffitt Cancer Center, Florida University of Chicago Medicine, Illinois Oregon Heath and Science University, Oregon Medical College of Wisconsin, Wisconsin United States	Recruiting

**Table 2 ijms-24-07201-t002:** Phase I and phase II CAR-T clinical trials for relapsed/refractory T-cell malignancy.

T-Cell Antigen	Clinical Trials	Phase	Condition	Age	Chemotherapy Prior to Infusion/Additional Intervention	Location	Status
CD 7	NCT04840875	I	HR acute T-cell leukemia/lymphoma	Up to 70 years old	Fludarabine, Cyclophosphamide	Beijing Boren Hospital, Beijing, China	Recruiting
	NCT04264078	Early Phase I	R/R T-cell leukemia/lymphoma	18–70 years old	Fludarabine, Cyclophosphamide and/or Melphalan	Xinqiao Hospital, Chongqing, China	Recruiting
	NCT04823091	I	R/R T-cell lymphoid malignancies	14–70 years old	Fludarabine, Cyclophosphamide	Union Hospital, Huazhong University of Science and Technology, Wuhan, China	Recruiting
	NCT04572308	I	R/R T-cell Acute Leukemia/lymphoma	2–65 years old	Fludarabine, Cyclophosphamide	Hebei yanda Ludaopei Hospital, Yanda, China	Completed
	NCT05397184	I	R/R T-cell Acute Lymphoid Leukemia	6 months–16 years old		Great Ormond Street Hospital for Children NHS Foundation Trust, London, United Kingdom	Recruiting
	NCT05290155	I	R/R T-Lymphoblastic Leukemia/Lymphoma/T-cell Acute Lymphoblastic Leukemia/Peripheral T-Cell Lymphoma/Angioimmunoblastic T-cell Lymphoma/Anaplastic Large Cell Lymphoma	14–70 years old		Shanghai General Hospital, Shanghai, China	Recruiting
	NCT05212584	I	R/R T-cell leukemia/lymphoma/High-Risk Hematologic Malignancies	2–60 years old		Hebei Yanda Lu Daopei Hospital, Langfang, China	Recruiting
	NCT05127135	I	R/R T-cell Acute Lymphoblastic Leukemia/Lymphoma/T-cell Non-Hodgkin Lymphoma	3–70 years old		The First Affiliated Hospital of USTC, Hefei, China Fundamenta Therapeutic Co. Ltd., Jiangsu, China	Recruiting
	NCT05043571	I	R/R T-cell Acute Lymphoblastic Leukemia/Lymphoblastic Leukemia	6 months–65 years old		National University Hospital, Singapore	Recruiting
	NCT04934774	I	R/R T-cell Acute Leukemia/lymphoma/T-cell Non-Hodgkin Lymphoma	18 years and older		Peking University Shenzhen Hospital, Guangdong, China	Recruiting
	NCT04480788	I	R/R CD7-positive hematological and lymphoid malignancies/T-cell Lymphoblastic Leukemia/Lymphoma	7–70 years old		The First Affiliated Hospital of Zhengzhou University, Zhengzhou, China	Recruiting
	NCT03690011	I	T-cell Acute Lymphoblastic Leukemia/Lymphoma/T-non-Hodgkin Lymphoma	Up to 75 years old		Houston Methodist Hospital, Houston, United States Texas Children’s Hospital, Houston, United States	Recruiting
	NCT04599556	I/II	R/R CD7+ T-cell Acute Lymphoblastic Leukemia/lymphoma/T-NHL/AML	3–80 years old		The First Affiliated Hospital, Zhejiang University, Hangzhou, China	Recruiting
	NCT04033302	I/II	CD7-positive hematological malignancies: T-cell Acute Lymphoblastic Leukemia/Lymphoma/AML/NK-Cell Lymphoma	6 months–75 years old		Shenzhen Geno-immune Medical Institute, Shenzhen, China	Recruiting
	NCT04762485	I/II	R/R CD7+ T-Lymphoblastic Leukemia/Lymphoma/Mixed Phenotype Acute Leukemia/AML	12–65 years old		The First Affiliated Hospital of Soochow University, Suzhou, China	Recruiting
	NCT04984356	I/II	R/R T-cell Acute Lymphoblastic Leukemia/Lymphoma	12 years and older	Fludarabine, Cyclophosphamide	City of Hope, Duarte, United States Children’s Hospital Los Angeles, Los Angeles, United States	Recruiting
	NCT04689659	II	R/R T-cell leukemia/lymphoma	1 to 70 years old		Beijing Boren Hospital, Beijing, China	Recruiting
CD4	NCT03829540	I	R/R T-cell leukemia/lymphoma	18 years and older		Indiana University Melvin and Bren Simon Comprehensive Cancer Center, Indianapolis, United States	Recruiting
CD5	NCT04594135	I	R/R T-cell Acute Lymphoblastic Leukemia/T-cell Non-Hodgkin Lymphoma	8 years and older		Peking University Shenzhen Hospital, Shenzhen, China	Recruiting
	NCT05487495	I	R/R T-cell Acute Lymphoblastic Leukemia/Lymphoma	1 to 70 years old	Fludarabine, Cyclophosphamide	Beijing Gaobo Boren Hospital, Beijing, China	Recruiting
	NCT05596266	I	R/R T-cell Acute lymphoblastic Leukemia	1–18 years old		Xuanwu Hospital Capital Medical University, Beijing, China	Recruiting
	NCT05032599	I	R/R T-cell Acute Lymphoblastic Leukemia	1–70 years old	Fludarabine, Cyclophosphamide	Beijing Boren Hospital, Beijing, China	Recruiting
Anti- TRBC1	NCT04828174	I	R/R TRBC1+ T-cell hematological malignancies/Acute T-Cell Leukemia	18–70 years old		Shanghai General Hospital, Shanghai, China	Recruiting
Anti-CD1a	NCT05745181	II	R/R acute T-Lymphoblastic Leukemia/Lymphoma	18–70 years old		The Affiliated Hospital of Xuzhou Medical University, Xuzhou, China	Recruiting

Abbreviations: R/R: relapsed/refractory.

## Data Availability

Not applicable.

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
