# Peer review of "An Update on Clinical Trials and Potential Therapeutic Strategies in T-Cell Acute Lymphoblastic Leukemia"

_ijms, 2023, doi:10.3390/ijms24087201_

Round 1
Reviewer 1 Report
In this review article the authors summarize the major therapeutic strategies and potential drug targets that are being evaluated in T-cell Acute lymphoblastic leukemia (T-ALL). The authors present a concise, well-structured revision and clear article.
The authors present the state of the art on the efficacy of T-ALL treatment protocols that is quite (fortunately) high. However, there are the cases of relapsed/refractory T-ALL patients that mostly occur in the adults were the toxicities associated with therapy decrease the survival rates, when compared to children.
The authors make an extensive list of the current treatment protocols that are being evaluated in clinical trials that show most promising preliminary results, but also other molecular targets that were tested without proven efficacy or it should be tested according to the authors opinion.
Overall, the article is very well written, structured, the ideas are presented clearly and well contextualized. The article is a great summarization of the ongoing promising therapies for this disease, and it will be well accepted in the T-ALL research community.
I only have a couple of minor issues with the article:
- - Line 122 – should be “have not shown” enhanced…” and not “have shown”. Reference 22 clearly states (corroborated by the data) that there is no enhanced cytotoxic effect in the combination of CDK4/6 inhibitors and other chemotherapeutic drugs.
- - In vivo and in vitro should be placed in italic throughout the text.
- - Line 158-160. Notch link should be explained here.
- - Line 275-276. I don’t understand the authors message, it seems that the sentence is not complete.
- - Line 300-303. This section refers to what? CAR-T therapy for T-ALL in general? Or specifically to CD99?
Reviewer 2 Report
Dear Authors,
Thank you for your contribution. I read your review on the new potential therapeutic strategies in T-ALL. I think it is a relevant topic in the oncology field where to date the survival is greatly increased, but the mortality remains high due to relapse, resistance to therapy or treatment-related toxicity. The aim should be finding new therapies able to target tumor specific antigens in order to avoid chemotherapy toxicity and to eradicate the tumor cells still remained after the first line therapy. In my opinion this is a quite complete review that gives the reader an overview of the progresses done and still to be done for patients with T-ALL.
My comments after reviewing are the following:
– line 53: please uniform the whole text putting the name of the drugs in capital letter or not
– line 55: I would reference the clinical trial AALL0434
– line 59: space before the title
– lines 72 and 74: I would add the p-value
– line 75: please explain the abbreviation CNS
– in general, when talking about a clinical trial, I would add the number of patients involved. For instance, 54% of patients achieved complete or partial response, so how many patients?
– section 2.1.2 on Bortezomib: it is not easy to follow. It seems not to talk about Bortezomib and I found a bit awkward the comparison between two different studies.
– line 108: I would put cyclin plural
– line 215: no space before anti-CD38
– line 215: I would cancel ‘anti-CD38 or’ because there are anti-CD38 antibodies other than Daratumumab
– line 221: you have used PDX before, please move the explanation of the abbreviation
– line 223: …showed promising results
– line 223: phase 2 II
– line 232: explain abbreviation AML
– line 236: explain abbreviation T-LBL
– line 237: phase 1 I
– line 253: I would add some numbers
– line 262: NCT number?
– line 275: this sentence is not easy to follow
– line 281: phase 2 II
– line 297: phase 1 I
– It is not clear to me why you wrote a separate section for CD99 CART-cells. I would keep all the CART in a section, which might end with your last sentence (line 300-303)
Kind regards
